# Diagnostic Yields and Clinical Impacts of Capsule Endoscopy

**DOI:** 10.3390/diagnostics11101842

**Published:** 2021-10-05

**Authors:** Seung Min Hong, Sung Hoon Jung, Dong Hoon Baek

**Affiliations:** 1Department of Internal Medicine, Pusan National University School of Medicine, Busan 49421, Korea; lucky77i@naver.com; 2Biomedical Research Institute, Pusan National University Hospital, Busan 49421, Korea; 3Department of Internal Medicine, Eunpyeong St. Mary’s Hospital, College of Medicine, The Catholic University of Korea, Seoul 03312, Korea; shjung74@catholic.ac.kr

**Keywords:** capsule endoscopy, diagnostic yields, clinical impact

## Abstract

Observing the entire small bowel is difficult due to the presence of complex loops and a long length. Capsule endoscopy (CE) provides a noninvasive and patient-friendly method for visualizing the small bowel and colon. Small bowel capsule endoscopy (SBCE) has a critical role in the diagnosis of small bowel disorders through the direct observation of the entire small bowel mucosa and is becoming the primary diagnostic tool for small bowel diseases. Recently, colon capsule endoscopy (CCE) was also considered safe and feasible for obtaining sufficient colonic images in patients with incomplete colonoscopy, in the absence of bowel obstruction. This review article assesses the current status of CE in terms of the diagnostic yield and the clinical impact of SBCE in patients with obscure gastrointestinal bleeding, who have known or suspected Crohn’s disease, small bowel tumor and inherited polyposis syndrome, celiac disease, and those who have undergone CCE.

## 1. Introduction

It is difficult to directly examine the small bowel because of its length (average length, 575 cm at the age of 20), and it is hard to use conventional endoscopic devices owing to the complex loops and length of the small bowel [1,2]. Conventional endoscopic devices can be used to observe only a small part of the proximal jejunum and distal ileum. Therefore, previously, small bowel radiography (SBR, small bowel follow-through (SBFT) and enteroclysis), push-enteroscopy, and small bowel angiography were used to examine the small bowel. The range of the small bowel that could be observed by push-enteroscopy was limited, despite considerable patient discomfort [3]. Therefore, the diagnostic yields of these imaging modalities were very low (push-enteroscopy, 35%, SBR, 37%) [4,5].

Small bowel capsule endoscopy (SBCE) provides a noninvasive and patient-friendly method for visualizing the small bowel. SBCE was developed by Gavriel Iddan in the mid-1990s and was introduced in 2000 [6]. Subsequently, double balloon enteroscopy (DBE) was launched in 2001 [7], and SBCE and DBE notably enhanced the diagnosis and treatment of small bowel diseases. With the ability to directly observe the small bowel using SBCE, the diagnostic yield for small bowel diseases and clinical physicians’ interest in treating small bowel diseases increased. Initially, after the development of SBCE, its diagnostic yield was higher than that of DBE. However, with more experience with DBE, there is almost no difference in the diagnostic yield in the two tests [8,9]. The reported pooled diagnostic yields for SBCE and DBE were 60% and 57%, respectively [9]. Although with SBCE, biopsy or therapeutic intervention is impossible and it is difficult to determine the exact location of the lesion, SBCE is the first choice for evaluation of small bowel diseases, owing to its convenience and safety [9].

The main indications for SBCE are obscure gastrointestinal bleeding (OGIB) and Crohn’s disease (CD). SBCE is also indicated for small bowel tumors and inherited polyposis syndromes, celiac disease, chronic diarrhea, and chronic abdominal pain, and for clarifying abnormal small intestinal findings from other imaging examinations. In previous research, OGIB accounted for 66% of all SBCE indications, and the remaining SBCE indications were distributed between chronic abdominal symptoms (10.6%), CD (10.4%), small bowel tumor (3.5%), and celiac disease (1.7%) [10,11]. Colon capsule endoscopy (CCE) has a high success rate (68 to 98%) in patients who underwent incomplete colonoscopy due to the severe pain, severe mesenteric redundancy, and postoperative adhesions [12,13,14,15,16]. Additionally, CCE can reduce the risk of colonoscopy complications such as perforation.

Since the introduction of capsule endoscopy (CE) in clinical practice in 2001, CE has been improved to include a wider field-of-view, high-resolution imaging, longer battery life, and miniaturization; thus, the clinical application of CE is gradually expanding. We evaluated the current status of CE in terms of its diagnostic yield and the clinical impact of SBCE on patients with gastrointestinal bleeding, who have known or suspected CD, small bowel tumor and inherited polyposis syndrome, celiac disease, and those who have undergone CCE.

## 2. Current Status of Small Bowel Capsule Endoscopy

Since Given’s first commercialization of SBCE in 2001, capsule endoscope systems currently in use include PillCam SB3^®^ (Medtronic, Dublin, Ireland), MiroCam^®^ (Intromedic, Seoul, Korea), CapsoCam^®^ (Capso-Vision, Saratoga, NY, USA), EndoCapsule^®^ (Olympus, Tokyo, Japan), and OMOM Capsule^®^ (Jinshan science and technology, Chongqing, China) (Table 1).

The PillCam SB^®^ (first generation; 11 × 26 mm) is equipped with one camera; it has a frame rate of 2 frames/s and a 140° field-of-view. The second and third-generation PillCam SB^®^ have the same size and number of cameras but the field-of-view is up to 156°. An adaptive frame rate system was added and compared to PillCam SB2^®^ (second generation), the resolution was improved by 30%; moreover, the operating time was extended to 11 h. In addition, it became possible to check the image of the capsule endoscope in real time through a real-time viewer. Intromedic (Seoul, South Korea) released MiroCam^®^ in 2007, and unlike other capsule endoscopes, this transmits information using human body communication, which reduces power consumption, and enables video recording for a longer period. The recently developed MiroCam MC 4000^®^ has two lenses at one side of the capsule that are displaced by approximately 4 mm, four LED lights, a wireless transmitter, and a battery. The depth range is up to 30 mm, the field-of-view is 170° and the frame rate is 2 × 2 frames/s. MiroCam MC 4000^®^ can perform size measurements and hardware-enabled 3D reconstruction using images from the two lenses. CapsoCam^®^, launched in 2013 by Capso-Vision of the United States, stores captured images in the capsule endoscope; thus, an external receiver and a data recorder are not required during the examination. However, this capsule endoscope must be found in the patient’s stool after the examination. In the CapsoCam design, four cameras were placed at 90° intervals in the middle of the capsule to provide a 360° field-of-view, and Capso-Cam SV-1^®^ can capture images at 12 to 20 frames/s. EndoCapsule^®^ (11 × 26 mm), released by Olympus (Tokyo, Japan) in 2005, captures images at 2 frames/s and has up to 145° field-of-view. The device self-adjusts its brightness and uses a high-resolution charge-coupled device lens to obtain an optimal image. The second-generation EndoCapsule^®^ has a wider field-of-view (160°) and has an operating time of 12 h, which is longer than that of the first generation EndoCapsule^®^. OMOM Capsule^®^, developed by Jinshan Science & Technology in China, has a size of 11 × 25.4 mm, and the second-generation OMOM Capsule^®^ has a wider field-of-view (165°) than the first generation, and the image quality is improved. This endoscope can capture images at 2 frames/s and has a battery life of about 10 h.

## 3. Diagnostic Yields of Small Bowel Capsule Endoscopy

Table 2 shows a summary of the diagnostic yields of SBCE.

### 3.1. Small Bowel Bleeding

OGIB is a commonly encountered clinical issue in gastroenterology and is associated with significant morbidity and mortality. OGIB originates in the small bowel in more than 80% of cases and is associated with vascular abnormalities, gastrointestinal tumors, and conditions such as Meckel diverticulum and CD [35]. Localization of small bowel bleeding is tedious for gastroenterologists due to the complex loops and long length of the small bowel. However, recent advances in imaging technology have changed the paradigm. Understanding the advantages and limitations, diagnostic yield, and therapeutic capabilities of these tests can help clinicians determine the most appropriate choice.

In the past, most cases of OGIB were not readily accessible for diagnosis and treatment, and occasionally, surgery or intraoperative endoscopy was required. However, with the active use of SBCE and balloon-assisted enteroscopy (BAE) in the recent years, it has become possible to identify the OGIB source in the gastrointestinal tract in most cases [36,37,38]. The diagnostic yield of SBCE for OGIB is 27–92.3% [17,18,19,20,21,22,23,24,25]. In three randomized controlled trials involving patients with OGIB and negative findings on esophagoduodenoscopy and ileocolonoscopy, the diagnostic yield of CE was significantly higher than those of SBR (27% vs. 4%; difference, 23%; 95% confidence interval [CI], 5–42) [26], angiography (53% vs. 20%; difference, 33%; 95% CI, 9–53; *p* = 0.016) [24], and push-enteroscopy (72.5% vs. 48.7%; *p* = 0.03) [25]. CTE (computed tomographic enterography) has a good diagnostic yield in the evaluation of patients with small bowel diseases. However, CTE is limited in its evaluation of OGIB and is relatively insensitive for small, infiltrative, flat, or inflammatory bleeding lesions of the small bowel [39]. According to a large-scale meta-analysis on CTE results for OGIB (18 studies; 660 patients), CTE showed a 40% (95% CI: 33–49) diagnostic yield for OGIB. In seven studies, the diagnostic yield of CTE vs. SBCE was 34% vs. 53% (95% CI: −34 to −4). In two studies (63 patients), the diagnostic yield of CTE vs. DBE was 38% vs. 78% (95% CI: −55 to −25) [40]. Other imaging techniques, such as radioisotope bleeding scan and angiography, were also insensitive in the absence of massive bleeding (rate of bleeding, ≥0.5 mL/min) [41,42].

In OGIB patients, the diagnostic yield of CE was similar to those of BAE and intraoperative enteroscopy [9,17,43]. However, BAE is invasive and time-consuming; general anesthesia is required in most cases, making it technically challenging to explore the entire small bowel. In contrast, SBCE is noninvasive, very well tolerated, and easy to perform [8,9]. In a meta-analysis on OGIB (10 published studies involving 651 patients), there was no statistically significant difference in diagnostic yield between CE and DBE (62% for CE vs. 56% for DBE; 95% CI, 0.64–2.29) [17]. Similar to the results of meta-analyses in the literature, the diagnostic yield for OGIB was 42.9–69% in Korean studies [18,19,20]. The diagnostic yield of DBE for CE-confirmed OGIB lesions was 75%, which was significantly higher than the yield when only DBE was performed (56%) (odds ratio, 1.79). Therefore, the diagnostic yield of DBE is better when DBE is performed after CE [17]. Thus, in a large group of patients with OGIB, DBE should be considered in a highly selected group, while CE can serve as a preliminary diagnostic tool in patients with small bowel bleeding.

The factor that most affects the diagnostic yield in OGIB is the bleeding status. The diagnostic yield is significantly increased when the examination is performed in the presence of active/occult bleeding [22,23]. Positive findings on CE were obtained in 92.3% of patients with ongoing overt bleeding, 12.9% in patients with previous overt bleeding, and 44.2% in patients with guaiac-positive stools and iron deficiency anemia [22]. In a large-scale retrospective cohort study, performing CE early after admission was associated with a higher diagnostic yield (55% on day 1, 48% on day 2, 29% on day 3, 27% on day 4, and 18% on day 5) [44]. According to Korean Gut Image Study Group (KASID) guidelines concerning OGIB published in 2013, CE is an effective initial diagnostic method for evaluating patients. Diagnostic yield is improved when CE is performed early in OGIB [45]. The 2015 European Society of Gastrointestinal Endoscopy (ESGE) guidelines also recommend SBCE be performed as soon as possible after a bleeding episode (within 14 days), in patients with OGIB [11]. Therefore, SBCE should be performed as early as possible after bleeding is identified. In addition, the diagnostic yield for OGIB was dependent on age, warfarin use, underlying liver disease, number of previous esophagogastroduodenoscopies, amount of blood transfusion required, and presence of connective tissue disease [46,47,48].

### 3.2. Small Bowel Tumor

Malignant neoplasms of the small bowel are among the rarest types of cancer, accounting for 1–2% of all gastrointestinal cancers [49]. Diagnosis of small bowel tumors via SBCE can be challenging. The clinical manifestations of small bowel tumors tend to be nonspecific, which can delay diagnosis, especially in the early stages. After the development of CE, the incidence of small bowel tumors increased to 2–10% [50,51].

In a recent retrospective study, the diagnostic yield of SBCE was 83.3% for small bowel tumors, whereas those for CT and SBFT were 55.8% and 46.1%, respectively. Sensitivity for detecting small bowel tumors was 40.4% for CT, 43.9% for SBFT, and 79.6% for SBCE [27]. Conclusively, it is considered that the diagnostic yield of SBCE is sufficient for the diagnosis and evaluation of small bowel tumors. However, the risk of false-negative SBCE results should always be considered, and this is more frequent for small bowel tumors and polyps in the duodenum and proximal jejunum, and submucosal tumors where a mucosal component is absent, such as gastrointestinal stromal tumors or neuroendocrine tumors [11]. In addition, the diagnosis rate of SBCE might be lower than that of CT or magnetic resonance imaging because SBCE cannot evaluate extraluminal status [52]. In this situation, magnetic resonance enterography (MRE) had higher specificity than SBCE, and CTE had higher sensitivity than SBCE [53,54]. Furthermore, the analysis of SBCE findings varies among interpreters; false-positive results are occasionally included. A scoring system that includes five reading components (bleeding, irregular surface, mucosal disruption, white villi, and color) of SBCE findings was proposed to overcome this shortcoming [55]. A prospective study evaluated SPICE (smooth, protruding lesion index on capsule endoscopy) score. The score has the following criteria for smooth protruding lesions: diameter larger than height, ill-defined boundary with the surrounding mucosa, non-visible lumen in the frames in which the lesion appears, and an image lasting less than 10 min.The score had a specificity of 89% and a sensitivity of 83% [56]. Further larger prospective studies are needed to validate such scoring systems.

Caution is required during examination since capsule retention may occur due to the presence of small bowel tumor. However, in the case of malignant tumors, capsule retention may serve as a marker of lesion location, which may be advantageous therapeutically [57].

### 3.3. Inherited Polyposis Syndrome

Intestinal polyposis syndromes are relatively rare. Intestinal polyposis syndromes can be divided, based on histology, into broad categories of familial adenomatous polyposis (FAP), hamartomatous polyposis syndromes (mainly including Peutz–Jeghers Syndrome (PJS), PTEN-associated hamartomatous syndromes, Cronkhite–Canada syndrome, and familial juvenile polyposis), and other rare polyposis syndromes such as serrated polyposis syndrome and hereditary mixed polyposis syndrome.

Prior to the development of SBCE, the diagnosis of small bowel polyposis was via SBR. It is difficult to access the tumor through a conventional endoscope when small bowel polyps are diagnosed using SBR. Thus, surgery or endoscopic polypectomy during surgery was performed. However, with the recent development of SBCE and BAE, SBCE is reported to have a higher sensitivity than SBR in diagnosing small bowel polyps in inherited polyposis syndrome [58]. The detection rate of SBCE for jejunal-ileal polyp is 24–30% in FAP patients and 90% (10 of 11 patients) in PJS patients [28,29]. Thus, small bowel polyp, which is difficult to access with a conventional endoscope, can be removed through BAE without surgery [59].

It is difficult to evaluate the duodenum, ampulla, and size of polyps through SBCE. A study reported that SBCE could detect duodenal polyps in only 36.4% of patients with endoscopically identified FAP [29]. Therefore, conventional endoscopic devices are recommended to evaluate the proximal small bowel in patients with FAP [11]. However, as more than 75% of patients with FAP and PJS have small bowel polyps and the risk of small bowel polyp increases with the presence of a duodenal polyp, SBCE can be considered when small bowel investigation is clinically required [28,60,61]. 

In a study by Burke et al., small bowel polyps were observed in 60% of FAP cases and 75% of PJS cases through SBCE examination, and the treatment plan was changed in 50% of patients. Therefore, the role of SBCE in detecting inherited polyposis syndrome is expanding [60]. However, it is important to note that occasionally, SBCE may miss a large polyp. Several studies have reported that MRE could detect large polyps (>15 mm) better than CE, and compared to CE, the result of MRE is more reproducible [60,62,63,64]. SBCE is recommended for small bowel surveillance in patients with polyposis syndrome, especially in patients with PJS, who are at high risk of intussusception and bleeding related to small bowel polyps [11,61].

### 3.4. Crohn’s Disease

CD is a chronic, progressive inflammatory bowel disease that can affect any segment of the gastrointestinal tract but commonly involves the small bowel in up to 60% of cases [65]. Small bowel CD is associated with serious complications such as stricture, abscess, and obstruction [66,67]. Small bowel CD has been underestimated due to diagnostic limitations in visualizing the small bowel [68,69]. CD is diagnosed by combining clinical features (abdominal pain or diarrhea for more than 6 weeks), laboratory test results (such as *C*-reactive protein level, fecal calprotectin level, and anemia or hypoalbuminemia), radiologic imaging, endoscopic evaluation, and histologic findings. Conventional diagnostic tools such as SBR, push-enteroscopy, and ileocolonoscopy have been used for small bowel CD, but these tools are limited by the difficult test procedure and the impossibility of detailed direct observation of intraluminal lesions. ESGE and Canadian guidelines suggest that CE is the initial diagnostic tool for assessing pathognomic symptoms of CD in the presence of a negative ileocolonoscopy examination and in the absence of obstructive symptoms or radiologic stenosis [11,61]. In addition, SBCE is recommended in patients with established CD, who have unclear clinical features on ileocolonoscopy or cross-sectional imaging, and in patients with established CD to confirm small intestinal mucosal healing.

In CD, examination of the terminal ileum during ileocolonoscopy may be important for diagnosis. However, the disadvantage of ileocolonoscopy is that only a part of the distal terminal ileum can be observed, and if the colon is stenosed, the scope cannot reach the cecum or intubation to the ileum. In addition, push-enteroscopy can be used to observe only 80–120 cm beyond the ligament of Treitz, and there are many complications: thus, there is a limit to its use [70]. SBFT is the most traditional method for obtaining images for small bowel evaluation in CD patients, but the patient is exposed to radiation and the diagnostic accuracy is related to the examiner’s experience [71]. In patients with suspected CD, the diagnostic yield of SBFT was only approximately 35%, whereas the diagnostic yield of SBCE was 70% [72]. Furthermore, although CTE has high diagnostic sensitivity, it cannot visualize lesions directly, whereas SBCE can visualize subtle changes in the entire small bowel mucosa. In other studies, in patients with a negative or inconclusive conventional workup (including ileocolonoscopy, CTE/MRE, or SBR) to diagnose CD, SBCE showed good sensitivity (93%) and specificity (84%) [73], and SBCE led to an incremental diagnostic yield of 24% [74]. Therefore, SBCE can confirm the presence of a lesion in the small intestinal mucosa in patients with suspected CD and in patients with known ileal and colonic CD; moreover, extensive small bowel involvement in CD can be evaluated.

According to a meta-analysis of 19 trials, when SBCE was performed in patients with suspected CD, its diagnostic yield was significantly higher than that of SBR, colonoscopy with ileoscopy (C + IL), and CTE (CE vs. SBR: 52% vs. 16 %; 95% CI, 16–48; *p* < 0.00001, CE vs. CTE: 68% vs. 21 %; 95% CI, 31–63; *p* = 0.009, CE vs. C + IL: 47% vs. 25%; 95% CI, 5–39; *p* < 0.00001). However, no statistical difference was found in diagnostic yield in the meta-analysis between CE and MRE in cases of suspected SBCD (CE, 55% vs. MRE, 45%; incremental yield, 10%; 95% CI, 14–34; *p* = 0.43) [30]. In the literature, compared to MRE, SBCE showed equal or superior diagnostic yield [31,75,76]. In addition, when the analysis was performed in a real-world setting, the diagnostic yield of SBCE for CD was 50% [32].

In a study by Solem et al., the sensitivity and specificity of SBFT, ileocolonoscopy, CTE, and SBCE were compared in 41 patients with suspected or established CD. Test accuracy was 86% for ileocolonoscopy, 85% for CTE, 79% for SBFT, and 67% for SBCE. When the two tests were combined, SBCE and other small bowel imaging tests showed the highest sensitivity (92–100%) [77]. In symptomatic patients with an established diagnosis of CD, the diagnostic yield of SBCE was good (approximately 50%) [33]. However, CTE is more useful than SBCE in more advanced CD, such as bowel obstruction, fistula, and abscess [78]. In a prospective study by Gralnek et al., SBCE often led to a definitive diagnosis of CD, and CE influenced decision making in 72% of patients and led to a change in management in 78% of patients [79]. 

In evaluating recurrence in patients with CD who underwent surgery, SBCE showed superior yield than ileocolonoscopy (62% vs. 25%), with the advantage of detecting proximal small bowel lesions. It is difficult to pass a surgical anastomosis and observe the proximal part by ileocolonoscopy in patients who underwent side-to-side reconstitution of a neoiluem, which is why CE is more useful [80].

Before applying SBCE in CD, it is necessary to note the stenotic characteristic of CD. In the case of the small bowel, there may be no symptoms of obstruction until the lumen is almost blocked, because easy passage of liquid substances is retained in the stenotic situation. Therefore, although the capsule endoscope is small, it can be retained in the stenosis for a long time or cause complete occlusion. If this obstruction requires open surgery, it will pose a great threat to patients with stenosis [81]. Even if the endoscope passes through the stenosis if it takes much time to pass the stenosed area, the distal part of the stenosis may not be visualized due to battery exhaustion. Another limitation of SBCE is that it can be difficult to locate small intestinal lesions, and it takes 1–2 h to read the input image properly [82]. However, despite these limitations, it seems clear that SBCE can be added as a new tool for evaluating small bowel involvement in CD. 

### 3.5. Celiac Disease

Celiac disease is an autoimmune, gluten-induced small bowel enteropathy. Anti-tissue transglutaminase and anti-endomysial antibodies are used as serological diagnostic markers for celiac disease. Evaluating an individual’s response to a gluten-free diet is also a diagnostic method. However, the most important diagnostic approach is to confirm the presence of villous atrophy in the duodenum and small bowel via esophagogastroduodenoscopic biopsy. According to a meta-analysis, the sensitivity of CE in celiac disease was 89% (95% CI, 82–94) and the specificity was 95% (95% CI, 89–98). However, when the patient tests positive for anti-tissue transglutaminase and anti-endomysial antibodies, the positive predictive value and specificity of the endoscopic markers for celiac disease are 100% [11]. In a case of suspected celiac disease, the diagnostic yield was approximately 54% [34], but in general, CE is not recommended for suspected celiac disease. However, if there is no response to the appropriate treatment, CE should be considered, to differentiate other etiologies and evaluate the complications of celiac disease. Particularly, it is important to differentiate refractory celiac disease because of the associated risk of intestinal T-cell lymphoma. CE had higher concordance than optic endoscopy for histology of villous atrophy in refractory celiac disease (κ coefficient = 0.45 vs. 0.24, *p* < 0.001), and extensive mucosal damage observed on CE was correlated with patient nutritional status (*p* = 0.003). Additionally, extensive mucosal damage observed on CE could predict the type of refractory celiac disease (refractory celiac disease type II) [83]. Conversely, refractory celiac disease type I showed low diagnostic yield in imaging procedures, including CE [84]. A review of the literature revealed that most studies related to celiac disease included a small number of subjects; thus, additional studies are needed in the future.

## 4. Current Status of Colon Capsule Endoscopy

In 2006, first-generation CCE was initially released by Given Imaging (PillCam^®^ COLON, Given Imaging (eventually purchased by Medtronic), Yokneam, Israel) [85]. CCE-1 (PillCam COLON^®^, first-generation; 11 × 31 mm in size) is equipped with two cameras; it has a frame rate of 4 frames/s and a 156° field-of-view on both sides. A prospective study with CCE-1 showed that the sensitivity of CCE-1 for detecting colonic lesions was low compared to colonoscopy and the results were unsatisfactory (low sensitivity for detection of colon polyps (64%), advanced adenomas (73%), and colorectal cancers (74%)) [86]. As a result, second-generation CCE-2 (PillCam COLON2^®^, second generation; 11.6 × 31.5 mm in size) was developed to achieve higher sensitivity in 2014. CCE-2 has two high-resolution cameras providing the field-of-view of 172° for each camera, allowing a nearly full visual coverage of the colonic mucosa [87]. It has been endowed with a battery lasting about 10 h and has an adaptive frame rate system (setting the frame rate to 4 or 35 frames/s). Thus, a prospective European multicenter study demonstrated that the detection rate of colon polyps of >5 mm using CCE-2 was almost equivalent to colonoscopy [88]. 

## 5. Diagnostic Yield of Colon Capsule Endoscopy

CCE is safe and feasible for obtaining sufficient colonic images without obstruction in patients who underwent incomplete colonoscopy [89]. Recent publications have indicated CCE investigations as feasible after incomplete colonoscopy [16,90,91]. Further, studies have indicated patients’ preference for CCE compared to colonoscopy [92,93], and the complication rates of CCE are very low [12,13,14,15,16]. The success rate of CCE is reported to be 68–98% [12,13,14,15,16]. Table 3 shows a summary of the diagnostic yield of CCE.

In a multicenter large-scale cohort study conducted in the Netherlands in 2005, 9.7% (511/5278 patients) of colonoscopy examinees underwent incomplete colonoscopy. Non-advanced adenoma was found in unexplored colon segments in 2.3% of patients, advanced adenoma in 0.8%, and colorectal cancer in 3.5% [96]. In a study on the diagnostic yield and relative sensitivity of CCE, compared to those of CT colonography (CTC), after incomplete colonoscopy, the diagnostic yield for polyps ≥5 mm for CCE vs. CTC was 41.24% (95% CI, 31.34–51.69) vs. 15.46% (95% CI, 8.92–24.22), and the relative sensitivity of CCE was 2.67 (95% CI, 1.76–4.04). Further, the diagnostic yield for polyps ≥ 9 mm for CCE was 21.65% (95% CI, 13.93–31.17), while that for CTC was 11.34% (95% CI, 5.80–19.39), and the relative sensitivity of CCE was 1.91 (95% CI, 1.18–3.09) [12]. In addition, in a prospective, comparative study that compared CCE and CTC in patients who underwent incomplete colonoscopy, the per-patient sensitivity of CCE compared to CTC was 2.0 (95% CI, 1.34–2.98) for polyps ≥6 mm and 1.67 (95% CI, 0.69–4.00) for larger polyps (≥10 mm) [16]. Therefore, the overall diagnostic yield of CCE was superior to that of CTC. Capsule retention may occur in 1.4–2% of cases, and the cause may include ileal stricture due to unknown CD without symptoms and inflammatory stricture related to non-steroidal anti-inflammatory drug use [14,90].

## 6. Clinical Impact

SBCE has an important clinical impact on further diagnostic workup, therapeutic strategy, and long-term clinical evaluation in patients with OGIB, with favorable outcomes. According to Katsinelos et al., in patients with OGIB, the management plan was changed, according to SBCE findings; SBCE findings in 45 out of 63 patients (71.43%) led to the introduction of a therapy that resolved the underlying disease or improved the clinical condition during long-term follow-up (11.8 ± 8.7 months) [97]. In another study, SBCE findings (66.3%, 61 of 92 patients) showed the absence of overt bleeding and a normal hemoglobin value, and a 100% resolution of OGIB in young adults (<50 years old) was observed in the long-term follow up (range 81–1348 days) [98]. A negative SBCE finding was associated with a low risk of recurrent bleeding, and a significant abnormal finding was an independent predictor of recurrent bleeding. (HR = 2.4, 95% CI, 1.1–5.8) [99,100]. Further, although SBCE showed negative findings related to OGIB, it detected other digestive lesions and had an indirect clinical impact [101]. Although there are cases of small bowel tumors causing OGIB (or iron deficiency anemia), there are also cases of small bowel mass or polyp causing OGIB, based on SBCE findings. Additionally, guidelines recommend early use of SBCE to evaluate small bowel tumors when the cause of OGIB and iron deficiency anemia is unknown [11].

SBCE can identify lesions compatible with suspected CD, with the consequent change in treatment options for patients. Usually, the diagnostic delay from the onset of CD to diagnosis can be up to 7 years [102,103]. In the past, differential diagnosis of CD from tuberculosis, celiac disease, or Bechet’s disease was very difficult, but the introduction of SBCE has been helpful in determining diagnosis. If differential diagnosis is difficult, follow-up SBCE may be performed to evaluate the healing status of the mucosa after treatment. SBCE which was performed to evaluate OGIB enables early diagnosis of CD, allowing early initiation of treatment. Moreover, since CD with bleeding is often in the active phase, it may be associated with the early use of biologics. Urgesi et al. reported that findings indicative of CD were confirmed in 19.1% (94 of 492 patients) of patients who underwent SBCE for suspected OGIB. Subsequently, the diagnosis was confirmed during follow-up among these patients in whom CD was detected via SBCE [104]. SBCE can alter management plans in up to a third of patients with CD [105]. The jejunal involvement in CD is associated with a significant risk of further clinical relapse, stricture disease, and multiple abdominal surgeries, and SBCE enables the determination of disease prognosis [106]. Thus, identifying the proximal small bowel via lesions detected on SBCE has enhanced medical management. Mucosal healing can be evaluated by SBCE to monitor the effect of medical treatment in patients with CD, with a significant correlation between the Lewis score and fecal calprotectin [107]. SBCE is also used to diagnose recurrences of CD after surgery, and CE might increase diagnostic accuracy and impact therapeutic decisions [80]. Finally, it is necessary to consider the active use of SBCE in the treatment and assessment of treatment response in patients with CD.

SBCE also has a significant impact on celiac disease. Since positive endoscopic SBCE finding in patients with positive endomysial antibody/transglutaminase antibody has high specificity [11], SBCE can be used as a good diagnostic tool in patients who are unable to undergo esophagogastroduodenoscopy [108]. In newly detected uncomplicated celiac disease, atrophy is usually confined to the proximal small bowel on SBCE, and this may indicate a favorable outcome [109]. In addition, SBCE can be used to assess complicated diseases, such as malignancy or ulceration, when the clinical course of the disease shows progressive worsening, even in the presence of treatment and a gluten-free diet. In this case, performing CTE/MRE with SBCE may be helpful [110].

When reviewing the clinical impact of CCE on medical decision making in patients who underwent incomplete colonoscopy, it was found that the treatment plan was changed in 5.7–56.6% of cases (colectomy, colorectal cancer operation, and therapeutic colonoscopy and polypectomy were performed in 0.9–1.3%, 2–5.7%, and 17.6–56.6% of cases, respectively) [12,13,14,15,16,94,95]. Further, in 5–15% of patients undergoing colonoscopy, the scope may not be able to reach the cecum due to the severe pain, severe mesenteric redundancy, and postoperative adhesions [111]. Therefore, in elderly individuals, women, and patients with a history of abdominal or pelvic surgery, in whom there is a likelihood of incomplete colonoscopy, CCE can be alternatively considered [112]. In addition, it is important for patients with FAP to undergo colorectal cancer screening at least every 2 years from the age of 10–12. An advantage of CCE is that it can reduce the risk of colonoscopy-related complications, such as perforation, and it can be performed in children and adolescents who are at high risk of developing colorectal cancer but refuse to undergo colonoscopy due to the associated discomfort [113].

## 7. Conclusions

Since the introduction of SBCE in clinical practice in 2001, SBCE has been improved to include a wider field-of-view, high-resolution imaging, longer battery life, and miniaturization; thus, the clinical application of CE is gradually expanding. SBCE has proven a high diagnostic yield and is often the preferred initial diagnostic test in patients with OGIB, who have known or suspected CD, small bowel tumor, and inherited polyposis syndrome because of its noninvasive quality, better tolerance, and ability to view the entire small bowel. CCE is also considered safe and feasible for obtaining sufficient colonic images in patients with incomplete colonoscopy, with possible use in colorectal screening.

CE is currently available only as a diagnostic tool and still has limitations; future capsule prototypes seem necessary. Three-dimensional reconstruction of high-resolution imaging, high-frame-rate imaging, full spherical imaging, and capsule chromoendoscopy can reduce unnecessary invasive examination by clarifying the endoscopic and microscopic characteristics of small bowel lesions [114,115]. Controlling the movement of the capsule endoscope enables complete observation of the small bowel mucosa (without blind spots), thereby increasing the diagnosis rate of lesions [116]. Furthermore, CE is considered a therapeutic endoscope for biopsies, hemostasis, and drug delivery [117]. In addition, using computer-assisted diagnosis and artificial intelligence, CE reading could be achieved, reducing reading time and improving diagnosis [118,119]. With the development of these technologies, it is expected to be used in clinical applications as actual products in the near future.

## Figures and Tables

**Table 1 diagnostics-11-01842-t001:** Specifications of the currently available capsule endoscopy systems.

Capsule Endoscopy	Manufacturer (Country)	Dimensions (mm)	Imaging Heads	Field of View (°)	Battery Life (h)	Frame Rate (fps)	Communication	FDA Approval
**PillCam SB3^®^**	Medtronic (USA)	11.4 × 26.4	1	156	≥11	2–6	RF	Yes
**MiroCam 4000^®^**	IntroMedic (South Korea)	10.8 × 24.5	2 (at one end)	170	10	2 × 2	EEP	Yes
**CapsoCam SV-1^®^**	CapsoVision (USA)	11 × 31	4	360	15	12–20	no communication (stored in capsule)	Yes
**Endocapsule 10^®^**	Olympus (Japan)	11 × 26	1	160	12	2	RF	Yes
**OMOM Capsule2^®^**	Jinshan Science and Technology (China)	11 × 25.4	1	165	10	2	RF	No
**PillCam Colon2^®^**	Medtronic (USA)	11.6 × 32.3	2	172 (per end)	10	4–35	RF	Yes

fps, frames per second; RF, radiofrequency; EFP, electric field propagation; FDA, Food and Drug Administration.

**Table 2 diagnostics-11-01842-t002:** Summary of diagnostic yield according to each indication of small bowel capsule endoscopy.

Author, Year [Ref]	Country	Study Design	Study Period	Number of Patients	Results
**Obscure Gastrointestinal Bleeding**					
Teshima, 2011 [17]	Netherlands	meta-analysis	N/A	10 trials (1239 patients)	Diagnostic yield: SBCE 61.7% vs. DBE 55.5% Diagnostic yield for DBE after a previously positive SBCE: 75.0% Diagnostic yield for DBE after a previously negative SBCE: 27.5%
Kim, 2005 [18]	Korea	retrospective	N/A	21	Diagnostic yield: 42.9%
Jang, 2007 [19]	Korea	retrospective	2003–2005	60	Diagnostic yield: SBCE 65.7% vs. DBE 80%
Lee, 2007 [20]	Korea	retrospective	2002–2004	126	Diagnostic yield: 69%
Ell, 2002 [21]	Germany	prospective	2001–2001	65	Diagnostic yield: SBCE 66% vs. push-enteroscopy 28%
Pennazio, 2004 [22]	Italy	prospective	2001–2002	100	Diagnostic yield - ongoing overt bleeding: 92.3% - previous overt bleeding: 12.9% - guaiac-positive stools and iron-deficiency anemia: 44.2%
Lepileur, 2012 [23]	France	retrospective	2004–2010	911	Diagnostic yield: 56%
Leung, 2012 [24]	China	prospective	2005–2007	60	Diagnostic yield: SBCE 53% vs. angiography 20%
Segarajasingam, 2015 [25]	Australia	prospective	2006–2009	79	Diagnostic yield: SBCE 72.5% vs. push-enteroscopy 48.7%
Laine, 2010 [26]	USA	RCT	2003–2008	136	Diagnostic yield: SBCE 27% vs. small bowel radiography 4%
**Small Bowel Tumor & Inherited Polyposis Syndrome**					
Han, 2015 [27]	Korea	retrospective	2004–2012	79	Diagnostic yields: CT 55.8% vs. SBFT 46.1% vs. SBCE 83.3% Sensitivity: CT 40.4%, vs. SBFT 43.9% vs. SBCE 79.6%
Schulmann, 2005 [28]	Germany	prospective	N/A	40	Diagnostic yield in patients with PJS: 90.9%
Iaquinto, 2008 [29]	Italy	prospective	N/A	23	Diagnostic yield in patients with FAP: 30.4%
**Crohn’s Disease**					
Dionisio, 2010 [30]	USA	meta-analysis	N/A	12 trials (428 patients)	Diagnostic yield for suspected Crohn’s disease -SBCE 52% vs. SBR 16% -SBCE 68% vs. CTE 21% -SBCE 47% vs. ileocolonoscopy 25% Diagnostic yield for established Crohn’s disease -SBCE 71% vs. SBR 36% -SBCE 66% vs. push-enterography 9% -SBCE 71% vs. CTE 39%
Jensen. 2011 [31]	Denmark	prospective	2007–2009	93	The sensitivity and specificity for diagnosis of Crohn’s disease of the TI -SBCE: 100% and 91% -MRE: 81% and 86% -CTE: 76% and 85%
Kharazmi, 2020 [32]	Denmark	retrospective	2011–2018	516	Diagnostic yield: 50%
Mehdizadeh, 2010 [33]	USA	retrospective	2001–2005	134	Diagnostic yield: 52%
**Celiac Disease**					
Lujan-Sanchis M, 2017 [34]	Spain	retrospective	2003–2015	163	Diagnostic yield: 54%

SBCE, small bowel capsule endoscopy; DBE, double balloon enteroscopy; SBR, small bowel radiography; CTE, computed tomographic enterography; MRE, magnetic resonance enterography; PJS, Peutz–Jeghers Syndrome; FAP, familial adenomatous polyposis; N/A, not applicable.

**Table 3 diagnostics-11-01842-t003:** Summary of diagnostic yield of colon capsule endoscopy.

Author, Year [Ref]	Country	Study Design	Study Period	Number of Patients	Results
Otani, 2020 [13]	Japan	retrospective	2011–2017	60	Diagnostic yield (for polyp larger than 6 mm): 56.7%
Hussey, 2018 [14]	Ireland	prospective	2015–2017	50	Diagnostic yield (new lesions after incomplete colonoscopy): 74%
Nogales, 2017 [15]	Spain	prospective	2010–2013	96	Diagnostic yield (new lesions after incomplete colonoscopy): 60.4%
Baltes, 2018 [90]	Germany	prospective	2010–2013	74	Diagnostic yield (for significant polyp * including adenocarcinoma): 28%
Negreanu, 2013 [93]	Romania	prospective	N/A	70	Diagnostic yield (any finding): 34%
Pioche, 2012 [94]	France	prospective	2008–2009	107	Diagnostic yield (new lesions after incomplete colonoscopy): 33.6%
Alarcon, 2013 [95]	Spain	prospective	2010–2011	34	Diagnostic yield (any finding): 34% Efficacy of CCE (allowing formulation of a specific medical plan): 58.8%

* Significant polyp, defined by size (≥6 mm) or number (≥3); N/A, not applicable.

## Data Availability

Not applicable.

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
