# Peer review of "Diagnostic Yields and Clinical Impacts of Capsule Endoscopy"

_diagnostics, 2021, doi:10.3390/diagnostics11101842_

Round 1
Reviewer 1 Report
This review evaluated data on capsule endoscopy for both small bowel and colon. The main technical characteristics of devices, as well as indications and results of capsule endoscopy studies according to different settings were reported.
The manuscript is updated, clearly written and clinically useful. As a gastroenterologist, I really enjoyed to read it, and I would really congratulate with the Authors.
I have not criticisms/comments to do.
Author Response
Thank you so much for reviewing our article.
Your comment was very helpful.
Thank you.

Reviewer 2 Report
Thank you for reviewing an article.
-Major comments-
1. Even considering that it is a review article, the contents and titles of "2. Current status ~ ("Subtitle 2") " and "Table 1" look very similar to the reference 18. This section must be revised.
2. In my opinion, the main purpose of this article is to summarize the diagnostic yield of SBCE and CCE for lower tract intestinal disorders. So, I think that modifications are necessary to improve the readability of "Table 2, Table 3".
-Minor comments-
1. There are a lot of references, and, the contents are vast. How about revising and compacting the duplicates or contents that can be omitted? (i.e; contraindication of SBCE in Introduction ; definition and classification of GI bleeding in 3.1 small bowel bleeding, similar contents in Introduction and Clinical impact,etc)
2. In "Abstract", how about unifying "small intestine and small bowel" / "large intestine and colon" into one word each? The meaning of the last two sentences is ambiguous. A little more clearer expression is needed. Also, in this article, there is no mention of magnetic controlled CE for gastric examination. So, the mention of Single CE is somewhat disconcerting.
3. In "4. Current status of Colon~"("Subtitle 4"), there is no content of CCE device unlike the previous paragraph ("Subtitle 2"). A brief mention of device seems necessary.
4. In "Clinical impact", there is no content on the clinical impact of CCE. This content seems to be in "Subtitle 4". The subtitle and the content should match.
5. In "Conclusion', the content of this paragraph seems to be related to the future prospective of CE rather than to summarize this article. A brief summary of the diagnostic yield and clinical impact in SBCE and CCE should be added.
Author Response
-Major comments-
- Even considering that it is a review article, the contents and titles of "2. Current status ~ ("Subtitle 2") " and "Table 1" look very similar to the reference 18. This section must be revised.
Response
Thank you for your comments. The contents of capsule endoscopy in Table 1 are summarized based on the brochures of each company. There are also many similarities with other capsule endoscopy review articles. We have put much effort into making this manuscript different from other articles. The contents of the revised manuscript are updated.
- In my opinion, the main purpose of this article is to summarize the diagnostic yield of SBCE and CCE for lower tract intestinal disorders. So, I think that modifications are necessary to improve the readability of "Table 2, Table 3".
Response
Thank you for your comments. The number of articles reviewed was too large, and as you have pointed out, there is too much content.
Therefore, Tables 2 and 3 were removed because it was thought that the corresponding texts were adequately explained.
-Minor comments-
- There are a lot of references, and, the contents are vast. How about revising and compacting the duplicates or contents that can be omitted? (i.e; contraindication of SBCE in Introduction ; definition and classification of GI bleeding in 3.1 small bowel bleeding, similar contents in Introduction and Clinical impact,etc)
Response
Thank you for your comments. We have tried to reduce the contents as much as possible.
- In "Abstract", how about unifying "small intestine and small bowel" / "large intestine and colon" into one word each? The meaning of the last two sentences is ambiguous. A little more clearer expression is needed. Also, in this article, there is no mention of magnetic controlled CE for gastric examination. So, the mention of Single CE is somewhat disconcerting.
Response
Thanks for the comments. We have unified the information about the SMALL INTESTINE and COLON as you have suggested.
- In "4. Current status of Colon~"("Subtitle 4"), there is no content of CCE device unlike the previous paragraph ("Subtitle 2"). A brief mention of device seems necessary.
Response
Thank you for your comments. Contents about CCE Device have been described in subtitle 4.
- In "Clinical impact", there is no content on the clinical impact of CCE. This content seems to be in "Subtitle 4". The subtitle and the content should match.
Response
Thanks for the comment. Contents about the Clinical impact of CCE have been described in the revised manuscript subtitle 5.
- In "Conclusion', the content of this paragraph seems to be related to the future prospective of CE rather than to summarize this article. A brief summary of the diagnostic yield and clinical impact in SBCE and CCE should be added.
Response
Thanks for your suggestion. A summary of the diagnostic yield and clinical impact in SBCE and CCE has been added to the manuscript.

Round 2
Reviewer 2 Report
Thank you for allowing me to review the revised manuscript. A great deal of contents is well organized.
- In the last sentence of the ABSTRACT, the meaning, especially “~small bowel tumor and inherited polyposis syndrome, celiac disease, and CCE”, is ambiguous. Please make the sentence clearer.
- On the line 18 of the INTRODUCTION, how about presenting the overall diagnostic yield for SBCE and DBE?
- 3. Diagnostic Yields -> 3. Diagnostic Yields of Small bowel Capsule Endoscopy?
- The title of ["Table 1. Specifications of the currently available capsule endoscopy systems." at your manuscript] and ["Table 1. Specification of currently available small-bowel capsule endoscopy systems" at reference] look very similar. Please review again.
- Instead of removing the TABLE 2 & 3, I suggested making it simpler the TABLE 2 & 3. Although well explained in the manuscript, I think it is very important to summarize the diagnostic yield by diseases (OGIB, CD, Polyp, etc) and/or tests (SBCE vs CTE, DBE. CCE etc) in this article. Please reconsider.
Author Response
2021.9.22.
Prof. Dr. Andreas Kjaer
Editor-in-Chief
Diagnostics
Dear Editor:
We/I wish to re-submit the manuscript titled “Diagnostic yields and clinical impacts of capsule endoscopy.”
We thank you and the reviewers for your thoughtful suggestions and insights. The manuscript has benefited from these insightful suggestions. I look forward to working with you and the reviewers to move this manuscript closer to publication in the Diagnostics.
The manuscript has been rechecked and the necessary changes have been made in accordance with the reviewers’ suggestions. The responses to all comments have been prepared and attached below.
Thank you for your consideration. I look forward to hearing from you.
Sincerely,
Dong Hoon Baek, MD, PhD,
Thank you so much for reviewing our article. Your comment was very helpful.
1. In the last sentence of the ABSTRACT, the meaning, especially “~small bowel tumor and inherited polyposis syndrome, celiac disease, and CCE”, is ambiguous. Please make the sentence clearer.
Thank you for your kind comments.
We have corrected the sentence as follow.
“This review article assessed the current status of CE, in terms of the diagnostic yield, and the clinical impact of SBCE in patients with obscure gastrointestinal bleeding, who have known or suspected Crohn’s disease, small bowel tumor and inherited polyposis syndrome, celiac disease, and those who have undergone CCE.”
Please let us know if this does not address your concerns.
2. On the line 18 of the INTRODUCTION, how about presenting the overall diagnostic yield for SBCE and DBE?
Thank you for your kind comments.
The following sentence has been inserted.
“The reported pooled diagnostic yield for SBCE and DBE was 60% and 57%, respectively.9”
3. Diagnostic Yields -> 3. Diagnostic Yields of Small bowel Capsule Endoscopy?
Thank you for your kind comments.
We have corrected the sentence.
4. The title of ["Table 1. Specifications of the currently available capsule endoscopy systems." at your manuscript] and ["Table 1. Specification of currently available small-bowel capsule endoscopy systems" at reference] look very similar. Please review again.
Thank you for your comments. The contents of the revised manuscript are updated. We have put much effort into making this manuscript different from other articles.
5. Instead of removing the TABLE 2 & 3, I suggested making it simpler the TABLE 2 & 3. Although well explained in the manuscript, I think it is very important to summarize the diagnostic yield by diseases (OGIB, CD, Polyp, etc) and/or tests (SBCE vs CTE, DBE. CCE etc) in this article. Please reconsider.
Thank you for your kind comments and thoughtful questions.
The TABLE 2 & 3 has been added.
